# Fruit and Vegetable Consumption during the COVID-19 Lockdown in Serbia: An Online Survey

**DOI:** 10.3390/foods13010125

**Published:** 2023-12-29

**Authors:** Dragana Ubiparip Samek, Renata Kovač, Lato Pezo, Jasna Mastilović, Aleksandra Bajić, Žarko Kevrešan

**Affiliations:** 1Institute of Food Technology, University of Novi Sad, 21000 Novi Sad, Serbia; renata.kovac@fins.uns.ac.rs (R.K.); aleksandra.bajic@fins.uns.ac.rs (A.B.); zarko.kevresan@fins.uns.ac.rs (Ž.K.); 2Institute of General and Physical Chemistry, University of Belgrade, 11000 Belgrade, Serbia; latopezo@yahoo.co.uk; 3BioSense Institute, University of Novi Sad, 21000 Novi Sad, Serbia; jasna.mastilovic@biosense.rs

**Keywords:** COVID-19, fruits, vegetables, consumers, structural equation modelling, theory of planned behavior

## Abstract

The global impact of the COVID-19 pandemic has significantly influenced individuals’ dietary choices and food-buying behavior. Present research specifically delves into the alterations in fruit and vegetable (F&V) consumption among consumers in Serbia during the pandemic. The study employs an extended model of the theory of planned behavior (TPB), incorporating the construct of consumers’ knowledge to comprehensively explore behavioral changes. Conducted in the first half of 2020, the study engaged 479 participants from Serbia, using an online questionnaire for data collection. Structural equation modelling was employed for a thorough analysis of the gathered responses. The main outcome revealed a noteworthy connection between consumers’ knowledge and their attitudes, subjective norms, and intentions regarding F&V consumption. Subjective norms significantly influenced consumers’ desire to increase F&V intake during and after the outbreak. These intentions aligned with actual behavior changes, indicating a shift toward higher F&V consumption. Consumer knowledge also notably shaped attitudes and behaviors regarding F&V intake. Notably, the TPB proved valuable in predicting and understanding these dietary shifts during global crises like the pandemic. These insights not only bolster healthier eating habits but also have broader implications for public health. Understanding consumer behavior dynamics during crises like COVID-19 is crucial for crafting effective strategies to promote overall human health and well-being.

## 1. Introduction

Global change caused by the COVID-19 pandemic has also reflected on people’s eating and food-purchasing habits. Recently published studies revealed that many people changed their eating behaviors, mostly in a harmful way to their health [1,2,3,4,5,6,7,8]. A healthy diet, rich in fruits and vegetables (F&V), accompanied with regular exercise has a major role in boosting our immunity and keeping it in a good shape to fight off virus infection [5]. Five to nine daily servings of F&V significantly increase the antibody response, which is necessary in fighting various diseases, including COVID-19 [5,9]. Still, Litton and Beavers [10] reported a general decrease in F&V consumption (fresh, frozen, and canned) since the pandemic started. Huber et al. [7] reported that COVID-19 pandemic lockdown significantly affected eating habits in young adults and changed their nutritional behavior. At first, the change was evident in the food amount, but with time, it included alterations in food composition and acquirement [7]. Di Renzo et al. [6] conducted a study in Italy and observed weight gain in almost half of the respondents during COVID-19 lockdown. Research also reported a slight increase in physical activity and higher fruit and vegetable procurement from farmers or organic purchasing groups. In addition, authors reported a slight reduction in the number of smokers among Italians [6]. Werneck at al. [8] conducted a study among the population in Brazil and reported that people who tend to have sedentary behaviors were more prone to unhealthy diet during the COVID-19 pandemic quarantine.

Not surprisingly, the COVID-19 situation strongly affected food markets, especially the market for F&V [10,11,12]. Thus, the availability of these groceries to the consumers has been altered, at times even limited, and became another contributing factor in reshaping daily diet. Furthermore, a new USDA ERS report [13] shows much larger and more persistent impacts of COVID-19 on international food security than expected. Food insecurity is defined as “reduced quality, variety, or desirability of diet” and may have a negative effect on health [10]. This phenomenon is linked to a national and household level, which becomes more evident in low- and middle-income countries [10]. Thus, some populations may be at a higher risk of experiencing significant negative health impacts caused by poor diet during the pandemic, especially when it comes to low intake of F&V [10,13,14].

In Serbia, fresh F&V intake had already been characterized as insufficient and inadequate, in spite of their well-known and documented nutritional importance and beneficiary health effects [15,16]. Following COVID-19 outbreak in Serbia, lifestyles substantially changed. People’s eating behavior and food acquisition, in general, shifted due to applied containment measures. We tried to assess the impact of coronavirus pandemic on fresh F&V consumption among residents of Serbia, shortly after the COVID-19 outbreak in this country. To the authors’ knowledge, there have not been reports that used SEM on TPB for exploring F&V consumption in general, and especially in relation to the COVID-19 pandemic. The obtained results are among the first reports referring to Serbian population lifestyle and eating habits in light of the COVID-19 pandemic. Moreover, present study may be useful in creating guidelines for the future nutritional interventions in supporting and improving the health status of Serbian and other populations at a greater risk of negative health impacts.

In 2020, SARS coronavirus 2 (SARS-CoV-2) took the world by a surprise and, beyond anyone’s expectations, imposed a new reality upon modern society. The virus primarily attacks the human respiratory system [1], causing atypical pneumonia known as COVID-19 [2]. Reports state that the severity and outcome of this disease are dependent on patient’s age. Hence, persons above the age of 65 and those with chronic diseases (diabetes, chronic lung disease, cancer, chronic kidney disease, obesity) are especially susceptible to developing severe forms of COVID-19 with complications [3]. In an attempt to stop or, at least, control coronavirus transmission and save lives, general lockdowns along with voluntary social distancing and self-isolation were adopted worldwide. These decisions forced people to make sudden and radical changes in their everyday lifestyles, regular schedules, routines and habits. At the same time, applied measures brought a global recession and major shifts in international economic and trade conditions [4].

Having in mind contemporary studies regarding F&V consumption in Europe [17], as well as in Serbia [15,16], the primary objective of the present research was to delve deeper into the intricacies of F&V consumption during COVID-19 and its correlation with the behavior of the participants in Serbia.

The initial hypothesis (H1) proposes that F&V consumption involves a complex interplay of multiple factors. Subsequent hypotheses suggest that the knowledge (K) affects various elements, namely: (H2) overall attention toward F&V consumption (A); (H3) subjective norm (SN); (H4), and personal behavior control (PBC). Furthermore, the intention of F&V consumption (I) is expected to be influenced by previous knowledge (K) (H5), attitude to consume F&V (H6), subjective norms (H7) and personal behavior control (H8). The last hypothesis (H9) refers to the influence of intention toward behavior of F&V consumption.

## 2. Materials and Methods

### 2.1. Study Background

In order to explain and test the main determinants of F&V consumption among consumers in Serbia, during the COVID-19 outbreak, the theory of planned behavior (TPB) was used as a theoretical framework as it is often applied in analyzing consumers’ behavior patterns [18]. Keeping in mind that this theory may not necessarily capture all predictors of more complex behavior, according to the Ajzen’s recommendations, the theory was expanded with additional construct of “knowledge” [19,20] (Figure 1). Different studies confirmed that TPB predictive power increases when adding knowledge because the greater knowledge somebody has, the more confident they will be when making decisions [21].

Therefore, this paper aims to check the usefulness of this theory in explaining how consumers’ F&V consumption behavior changed during the COVID-19 pandemic. It also examines the role of consumers’ knowledge in creating targeted interventions to increase F&V consumption.

### 2.2. Survey Design and Data Collection

The cross-sectional study was conducted using an online questionnaire survey created in free online software via the Google^®^ platform (Google Forms, Alphabet, Mountain View, CA, USA). and was distributed all over Serbia. The invitation letter and the consent form were delivered to all participants. They were informed about the objectives of this research, and all of them gave their consent for participation in the research and publication of the obtained results in a scientific journal. The first step in creating the questionnaire was the development of the questions, based on the recommendations in [19]. Next, the content was validated by five experts who gave their opinion on the current COVID-19 pandemic situation in Serbia. Subsequently, the pilot test was performed among 20 consumers. The questionnaire consisted of two parts: a socio-demographic part, with 11 questions, and a TPB part, which included the statements about each TPB construct (Figure 1): consumers’ knowledge about F&V (4 statements), their attitudes towards F&V consumption (10 statements), subjective norms (4 statements), perceived behavioral control (6 statements), consumers’ intentions to increase F&V consumption (4 statements) and consumers’ behavior (4 statements).

After checking the validity and reliability of the TPB constructs, two statements related to attitudes were excluded. The statements used in the survey are presented in groups in Table 1.

Using non-probability sampling, participants were invited to take part in the survey by sharing the questionnaire link via emails and on different social-media platforms like Facebook, Instagram, Viber, and WhatsApp. The final sample consisted of 479 consumers who answered the questionnaire between 4 May and 16 June 2020 during the COVID-19 outbreak in Serbia in the period of strict lockdown for the citizens.

### 2.3. Sample Description

The sample consisted of 27.8% males and 72.2% females (Table 1). This disproportion can be explained by the fact that women are more engaged and influential on purchasing and creating family diet [22]. Additionally, the proportion of respondents aged between 25 and 45 years (63.0%), and of residents of big cities (59.7%) compared to those from villages or small towns around the country, was dominant. Moreover, 69.7% of respondents had an elementary or high school education level, whereas only 30.3% had a college degree. Moreover, 75.2% were employed while 76.8% had more than the average monthly income of approximately EUR 500. Analysis of the family structure showed that 63.0% of respondents have 3 or more family members. 

When asked about the frequency of meal preparation for other family members, 39.7% of respondents stated they make meals every day while 25.7% of them do that often. Consequently, 56.8% of consumers go grocery shopping often and 23% do that every day. As for the frequency of regular physical activity, 38.2% of respondents carried out physical activity a few times a week, while 22.8% of them had physical activity every day. The majority of respondents estimated their health condition as good (73.9%).

### 2.4. Measures and Statistical Analysis

The questionnaire items were measured according to [19]. TPB constructs were measured on 5-point Likert scale (1 denotes “strongly disagree” and 5 “strongly agree”) (Table 2) [23]. The data were analyzed to confirm the correlational relationships between knowledge, attitudes, subjective norms and perceived behavioral control with consumers’ intentions and F&V consumption behavior.

In order to investigate the predictors of consumers’ F&V consumption behavior during the COVID-19 pandemic and to test the explanatory power of TPB, the structural equation modelling (SEM) was used. SPSS 21.0 (IBM Corp., Armonk, NY, USA), Microsoft Excel 2010 (Microsoft Corporation, St. Redmond, WA, USA) and Ωnyx (Version 1.0-1026) were used for creating the presented research model. 

Residual statistics were used to check SEM requirements fulfilment. Collinearity statistics were tested through the variance inflation factors (VIF < 10) and tolerance (above 0.01) [24]. Descriptive statistics were used to check the variance for each TPB variable. The sample size was tested according to the calculation: http://danielsoper.com/statisticalc/calculator.aspx?id=89 (accessed on 1 March 2021), and it was confirmed that the minimum sample should be 256 survey members. Additionally, exploratory factor analysis (EFA) was used to test the adequacy of sampling through the Kaiser–Mayer–Olkin measure of sampling (KMO ≥ 0.5) [25,26], while the strength of the relationship among variables was assessed through Bartlett’s test of sphericity, where a value less than 0.05 indicates that these data should not be acceptable for further analysis [27]. The construct reliability was tested based on Cronbach’s alpha (≥0.7) [28]. Additionally, a rotated component matrix was created using principal component analysis (PCA) and the rotation method Varimax with Keiser normalization. Rotation converged in 7 iterations. 

Confirmatory Factor Analysis (CFA) was performed to check the validity and reliability of the measurement model, and the model fit was tested using the following fit indices: (Adjusted) Goodness of Fit ((A)GFI) as the proportion of variance accounted for the estimated population covariance (GFI ≥ 0.95; AGFI ≥ 0.90); Root Mean Square Error of Approximation (RMSEA) where values closer to 0 represent a good fit (RMSEA); Comparative Fit Index (CFI) that compares the fit of a target model to the fit of an independent model (CFI ≥ 0.90) [28].

## 3. Results

### 3.1. TPB Construct Relationships with Consumers’ Intentions and Consumption Behavior

When analyzing consumers’ knowledge about F&V consumption, over 90% of respondents agree that regular F&V consumption can improve general health (Table 2). Around 30% of them have a neutral opinion on whether regular F&V consumption can help in COVID-19 prevention whereas approximately 55% of them agree about their contribution to virus prevention. 

Respondents’ attitudes emphasized that over 90% of consumers consider F&V as high-quality groceries that are healthy and easy to use. When asked about their price, half of the consumers have a neutral opinion, but the majority of them agree that these groceries were more expensive during the outbreak (Table 2).

Interestingly, regarding subjective norms, more that 50% of respondents claimed that other people’s opinions and attitudes (including those from experts) are not crucial for their dietary behavior towards F&V consumption (Table 2).

The perceived behavioral control states the difficulties consumers face regarding F&V consumption. Over 70% of consumers claimed that F&Vs are not difficult to prepare for consumption. Moreover, over 60% of them did not face difficulties in buying F&V during the COVID-19 pandemic. Additionally, approximately half of them do not consider F&V as being dangerous and a source of infection by retaining the virus on their surfaces (Table 2). 

Results related to consumers’ intentions to increase F&V consumption show that 43.0% of respondents tried to increase their fruit consumption during the outbreak and 49.7% tried to increase their vegetable consumption, while approximately 30% had neutral intentions for both fruits and vegetables. When asked about their plans to increase the consumption of these groceries after the outbreak, 53% of respondents had plans to increase fruit consumption, while 53.9% of them planned the same for vegetables (Table 2). 

When asked about fruit consumption behavior, 38.6% of respondents usually eat one while 33.6% of them eat two fruits per day. The COVID-19 pandemic did not dramatically impact this frequency; hence, 35.1% of them ate one and 33.4 ate two fruits per day. Similar results were obtained for vegetable consumption behavior, since 45.1% eat one while 35.9 eat two vegetables per day. During the COVID-19 outbreak, one vegetable per day ate 41.1% of respondents, and 36.3% of them ate two pieces of vegetables (Table 2).

### 3.2. SEM Requirement Fulfilment and Model Testing

The results confirmed the fulfilment of all SEM requirements. Multivariate normality was checked and the residual statistics confirmed the model adequacy (Mahalanobis distance value of 201.05). The assumption of multicollinearity was not violated (VIF = 9.00) and tolerance as a measure of collinearity was 0.11 [24]. Descriptive statistics showed the variance of 2.04 and the recommended minimum sample size of 256 responses with 12 latent variables and 32 observed variables. EFA confirmed the adequacy of sampling through KMO = 0.70) [25,26], and the strength of the relationship among variables is confirmed by Bartlett’s test with a value higher than 0.05 [27].

Cronbach’s alpha for the construct consumers’ knowledge about F&V for all four statements included in the construct is 0.78; for consumers’ attitudes towards F&V for 8 statements, it is 0.70; for subjective norms for 4 statements, it is 0.92, for perceived behavioral control for 6 statements, it is 0.72; for 4 statements included in the construct intentions, it is 0.95; and for 4 statements that form construct behavior, it is 0.77. Of note, the statements AF3 (I consider fruits difficult/easy to use) and AV3 (I consider vegetables difficult/easy to use) do not have confirmed reliability, so they were excluded from model construction. As all the other values range from 0.70 to 0.95 [28], it can be confirmed that all the constructs are reliable and that the suggested model can be used to explain the F&V consumption among consumers in Serbia during COVID-19 outbreak. Results for rotated component matrix are presented in Table 3.

### 3.3. Model Identification

Model identification tested factor loading between the constructs and variables. The results in structural path coefficients are shown in Table 4 and structural path coefficients between constructs are provided in Table 5. 

The CFA results show the validity and reliability of the model, based on the results of fit indices used: GFI = 0.97, AGFI = 0.93, RMSEA = 0.93 and CFI = 0.87 [29], which confirm the utility of the TPB and presented model in explaining F&V consumption among consumers. The final SEM model based on TPB, as a theoretical framework, explaining consumers’ behavior related to F&V consumption during the COVID-19 outbreak in Serbia is presented in Figure 2.

## 4. Discussion

The present study examined the influence of behavioral predictors on F&V consumption among consumers in Serbia during the COVID-19 outbreak, by applying a modified TPB model, extended for the construct of knowledge. 

According to the results, a certain level of consumer’s knowledge about F&V nutritive aspects along with its beneficial health impact is necessary in order to increase the consumption of these groceries. During the period of the COVID-19 pandemic, the level of consumer knowledge has been affected by media exposure [21]. Serbian consumers have a high level of knowledge about F&V, considering it as a source of essential nutrients that improve health and support prevention of common non-communicable diseases [30]. That initial knowledge, supported by numerous findings in relevant scientific and medical literature, contributed to a high level of awareness about contribution of these groceries in COVID-19 prevention, which was subsequently confirmed by Moreb et al. [31]. If affected by the virus, adequate F&V consumption has an important role in boosting the immune system, in order to reduce the consequences of serious illnesses [31]. Knowledge, as extended construct, has a significant impact on consumers’ attitudes, subjective norms and their intentions to increase F&V consumption. On the other hand, it has no impact on perceived behavioral control. There are opposite results about the impact of knowledge. According to Mihalache et al. [32], knowledge impacts attitudes, while Da Cunha [33] opposes this finding. However, consumers’ knowledge alone, although it is necessary, is not enough to cause behavioral changes among consumers; thus, it must be related with their attitudes and subjective norms [34].

Serbian consumers have positive attitudes towards F&V consumption as they perceive them as high quality and healthy food. Contrary to Mucinhato et al. [21], where attitude presents the predictor with the strongest effect on intentions, in this study, attitude does not have a significant impact on consumers’ intentions, as found by Fulham and Mullan [35].

Subjective norms had a limited impact on consumers’ intentions to increase F&V consumption during the COVID-19 outbreak, as less than 50% of respondents claimed that they were encouraged by close people, experts and doctors. In a study conducted by Mucinhato et al. [21], subjective norms have a positive effect, but they are not the strongest predictor of intentions, as social pressure from others has an impact, but to a lesser extent. Being an important topic of public interest, the relationship between health and F&V consumption is frequently discussed in the media by experts, so the consumers can shape their opinion according to the social impact [21].

Perceived behavioral control, like attitudes, does not have an important impact on intentions, similarly to Lehberger [36]. In this study, perceived behavioral control did not have a significant impact on consumers’ intentions to increase F&V consumption, as consumers did not face limiting factors, such as difficulties to buy F&V or the risk of transferring the virus from the surface of these groceries. The present finding is not consistent with those of Fulham and Mullan [35] and Mucinhato et al. [21], who reported that perceived behavioral control has a significant impact on intentions. However, it is necessary to emphasize the importance of washing F&V before consumption, as viruses can survive on F&V surfaces and packages, if handled by an infected person [37]. On the other hand, although the F&V supply during the pandemic did not affect the majority of consumers, there are evident changes in F&V buying patterns. A lot of consumers moved to online grocery shopping from home [38]. As consumers mostly stocked non-perishable foodstuffs, they rapidly substituted fresh F&V for frozen and processed [12]. 

Finally, the study confirmed the important impact of consumers’ intentions on increasing F&V consumption after the COVID-19 outbreak. Eating behavior during the lockdown caused changes in eating behavior due to different factors, due to a restricted movement, loss of income, COVID-19 anxiety and others [39]. The change in purchasing frequency, caused by the coronavirus pandemic, led to a decrease in buying perishable foodstuffs, like F&V, which finally decreased the consumption of these groceries [40].

The practical use of this study was to examine the statistical viability of the hypotheses stated in Table 5, to be used for similar purposes to predict and understand human behavior and habits in unexpected circumstances. However, it is important to note that the main limitation is related to sampling and sample structure, since the majority of the participants were women. Therefore, in future studies, where possible and applicable, there should be a more balanced gender representation. Generally, sampling limitations in survey analysis include biases (sampling, non-response), small sample sizes, inaccuracies in sampling frames/methods, and challenges in generalizing findings. Addressing these concerns involves using appropriate techniques, ensuring diversity, minimizing non-response, and maintaining transparency in data collection.

Additionally, a significant contribution of this study lies in applying similar analyzes in other countries of the region, especially those in the Mediterranean area, since consumers have similar attitudes and habits towards F&V consumption.

## 5. Conclusions

This study was based on the TPB model, extended for the knowledge, in order to predict F&V consumption behavior during the COVID-19 outbreak in Serbia. SEM was used to explore the validity and reliability of TPB constructs in testing the key determinants of F&V consumption in COVID-19 context. The results confirmed the usefulness of tested model and showed that consumers’ knowledge about F&V in COVID-19 prevention has important role in increasing the consumption of these groceries, but knowledge alone is not enough to change the behavioral patterns. It is necessary to influence consumers’ attitudes and subjective norms in order to affect their intentions in increasing F&V consumption, which will, finally, result in specific behavior. This study emphasizes the importance of creating contemporary programs, measures and policies that will promote F&V beneficial health effects and strengthen consumers’ positive attitudes towards these groceries, which will ultimately change the consumption patterns of fresh F&V and help in fighting the ongoing pandemic.

## Figures and Tables

**Figure 1 foods-13-00125-f001:**
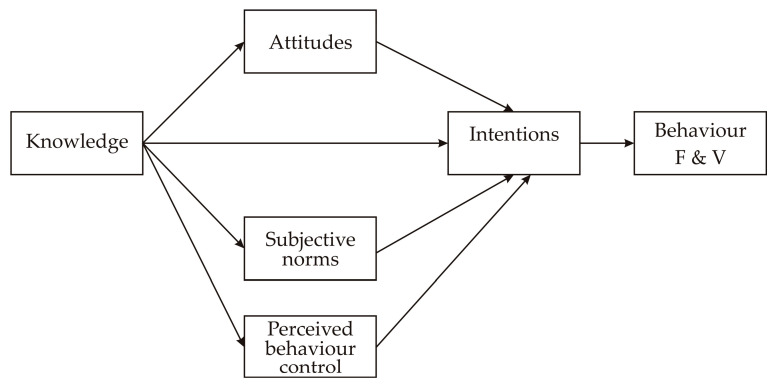
TPB model extended for “knowledge”, as an additional construct.

**Figure 2 foods-13-00125-f002:**
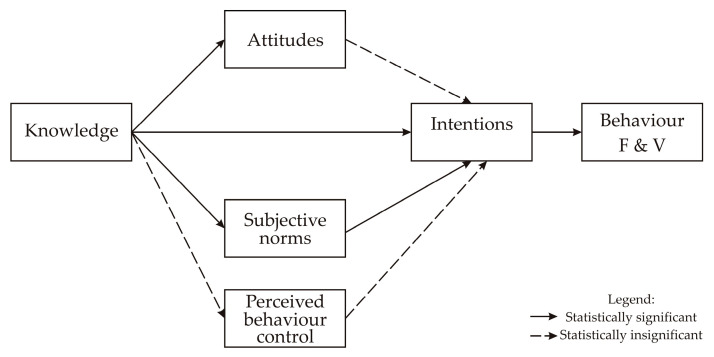
Final structural equation model TPB related to F&V consumption during COVID-19 outbreak in Serbia.

**Table 1 foods-13-00125-t001:** Socio-demographic characteristics of respondents.

Question	Alternatives	% of Respondents
Gender	male	27.8
	female	72.2
Age	up to 25 years	12.7
	26–45 years old	63.5
	over 45 years	23.8
Place of living	village	21.3
	town	19.0
	city	59.7
Education	elementary/high school	30.3
	undergraduate/graduated	69.7
Employment	student	9.0
	employed	75.2
	unemployed	10.8
	retired	5.0
Household members	one	10.3
	two	26.7
	three or more	63.0
Household monthly income	up to 50,000 dinars	23.2
	over 50,000 dinars	76.8
How often do you prepare meals for other family members	never	4.7
	rarely	11.9
	sometimes	18.0
	often	25.7
	every day	39.7
How often do you go grocery shopping?	never	0.6
	rarely	4.8
	sometimes	14.8
	often	56.8
	every day	23.0
Physical activity frequency	never	0.8
	rarely	20.9
	few times a month	17.3
	few times a week	38.2
	every day	22.8
Health condition self-estimation	bad	4.4
	good	73.9
	excellent	21.7

**Table 2 foods-13-00125-t002:** TPB constructs and percentage of responses.

TPB Constructs and % of Responses			1	2	3	4	5
	Regular Fruits Consumption:						
**Knowledge** **F & V (K)**	KF1	can contribute to better general health	0.8	1.5	8.1	**17.5**	**72.0**
	KF2	can help in COVID-19 prevention	8.8	7.9	**29.4**	**20**	**33.8**
	Regular vegetables consumption:						
	KV1	can contribute to better general health	0.2	0.8	6.5	**16.1**	**76.4**
	KV2	can help in COVID-19 prevention	8.1	8.1	**28.0**	**20.3**	**35.5**
	I consider fruits:						
**Attitudes F & V** **(A)**	AF1	low quality–high quality	0.6	0.6	4.0	**21.1**	**73.7**
	AF2	unhealthy–healthy	0.0	0.4	2.3	**17.3**	**80.0**
	AF3	difficult/easy to use	0.4	0.4	2.9	**10.9**	**85.4**
	AF4	expensive/cheap groceries	9.0	**15.7**	**55.7**	**13.2**	6.5
	AF5	more expensive/cheaper during the outbreak	**35.0**	**22.3**	**36.3**	4.3	2.1
	I consider vegetables:						
	AV1	low quality–high quality	0.0	0.2	4.0	**11.9**	**83.9**
	AV2	unhealthy–healthy	0.2	0.0	2.3	**9.4**	**88.1**
	AV3	difficult/easy to use	0.8	2.9	**18.2**	**23.4**	**54.7**
	AV4	expensive/cheap groceries	**9.2**	**11.5**	**52.2**	**16.5**	**10.6**
	AV5	more expensive/cheaper during the outbreak	**25.3**	**23.6**	**45.3**	4.1	1.7
**Subjective norms** **F & V** **(SN)**	SNF1	Close people, whose opinion is important to me encouraged me to consume more fruits during COVID-19 outbreak.	21.7	11.1	26.5	16.5	24.2
	SNF2	Experts and doctors encouraged me to consume more fruits during COVID-19 outbreak.	22.8	11.1	28.4	15.7	22.1
	SNV1	Close people, whose opinion is important to me encouraged me to consume more vegetables during COVID-19 outbreak.	21.5	10.4	28.8	15.9	23.4
	SNV2	Experts and doctors encouraged me to consume more vegetables during COVID-19 outbreak.	24.0	11.7	27.1	15.7	21.5
**Perceived behavioral control F & V** **(PBC)**	PBCF1	I find difficult to prepare fruits for consumption.	**69.5**	**11.7**	9.6	5.6	3.5
	PBCF2	During COVID-19 outbreak, I found difficulties to buy fruits.	**41.5**	**18.2**	17.5	15.0	7.7
	PBCF3	Fruits can cause COVID-19 by retaining on the surface.	**37.0**	**19.0**	**28.0**	7.5	8.6
	PBCV1	I find difficult to prepare vegetables for consumption.	**58.0**	**16.7**	16.9	5.0	3.3
	PBCV2	During COVID-19 outbreak, I found difficulties to buy vegetables.	**42.4**	**21.7**	18.2	11.9	5.8
	PBVC3	Vegetables can cause COVID-19 by retaining on the surface.	**37.8**	**19.8**	25.9	8.1	8.4
**Intentions F & V** **(I)**	IF1	I tried to increase fruits consumption during COVID-19 outbreak.	13.4	9.8	**33.8**	**17.5**	**25.5**
	IF2	I am planning to increase fruit consumption after COVID-19 outbreak.	11.3	8.8	**26.9**	**19.6**	**33.4**
	IV1	I tried to increase vegetables consumption during COVID-19 outbreak.	10.9	8.1	**31.3**	**22.8**	**26.9**
	IV2	I am planning to increase fruit consumption after COVID-19 outbreak.	10.4	6.9	**28.8**	**20.7**	**33.2**
			none	1	**2**	**3**	**>3**
**Behavior F & V** **(B)**	BF1	How many fruits do you usually eat per day?	7.3	**38.6**	**33.6**	10.2	10.2
	BF2	How many fruits did you eat during tCOVID-19 outbreak per day?	5.8	**35.1**	**33.4**	12.7	12.7
	BV1	How many vegetables do you usually eat per day?	3.1	**45.1**	**35.9**	11.9	4.0
	BV2	How many fruits did you eat during COVID-19 outbreak per day?	2.5	**41.1**	**36.3**	15.9	4.2

Likert scale: 1—totally disagree, 2—disagree, 3—neutral, 4—agree and 5—totally agree [23]; Bold numbers present significant values; For results interpretation results for 1 and 2 were analyzed together and show disagreement while results from 4 to 5 present agreement with presented statements.

**Table 3 foods-13-00125-t003:** Exploratory factor analysis (EFA)—rotated component matrix.

	Component										
	1	2	3	4	5	6	7	8	9	10	11
IF1	0.852										
IF2	0.887										
IV1	0.902										
IV2	0.908										
SNF1		0.841									
SNF2		0.876									
SNV1		0.864									
SNV2		0.867									
AF1			0.738								
AF2			0.773								
AV1			0.844								
AV2			0.849								
AF4				0.703							
AF5				0.803							
AV4				0.731							
AV5				0.854							
AF3					0.712						
AV3					0.687						
PBCF1					−0.690						
PBCV1					−0.778						
PBCF3						0.964					
PBCV3						0.970					
KF2							0.910				
KV2							0.920				
BV1								0.903			
BV2								0.920			
BF1									0.910		
BF2									0.916		
PBCF2										0.912	
PBCV2										0.902	
KF1											0.855
KV1											0.813

Extraction method: principal component analysis. Rotation method: Varimax with Kaiser normalization; rotation converged in 7 iterations.

**Table 4 foods-13-00125-t004:** Regression.

			Estimate	S.E.	C.R.	*p*
A	←	K	0.002	0.038	0.047	0.962
SN	←	K	0.321	0.056	5.743	***
PBC	←	K	0.043	0.052	0.838	0.402
I	←	K	0.245	0.046	5.371	***
I	←	A	−0.071	0.063	−1.114	0.265
I	←	SN	0.356	0.04	8.858	***
I	←	PBC	0.016	0.042	0.373	0.709
B	←	I	0.118	0.043	2.739	0.006
KF1	←	K	0.21	0.027	7.865	***
KF2	←	K	1			
KV1	←	K	0.197	0.025	7.939	***
KV2	←	K	0.995	0.036	27.288	***
AV4	←	A	0.669	0.06	11.145	***
AV2	←	A	−0.006	0.022	−0.278	0.781
AV1	←	A	0.02	0.026	0.746	0.456
AF5	←	A	0.889	0.062	14.445	***
AF4	←	A	0.524	0.056	9.398	***
AF2	←	A	0.016	0.027	0.594	0.553
AF1	←	A	0.037	0.032	1.16	0.246
SNF1	←	SN	0.727	0.04	18.384	***
SNF2	←	SN	1			
SNV1	←	SN	0.778	0.036	21.367	***
SNV2	←	SN	1.003	0.025	39.362	***
PBCF1	←	PBC	0.13	0.042	3.124	0.002
PBCF2	←	PBC	0.179	0.052	3.454	***
PBCF3	←	PBC	1			
PBCV1	←	PBC	0.186	0.04	4.617	***
PBCV2	←	PBC	0.279	0.049	5.694	***
PBCV3	←	PBC	1	0.044	22.606	***
IV2	←	I	0.981	0.029	34.141	***
IV1	←	I	1			
IF2	←	I	0.981	0.032	30.962	***
IF1	←	I	0.967	0.032	30.512	***
BF1	←	B	1			
BF2	←	B	0.972	0.064	15.212	***
BV1	←	B	0.283	0.042	6.698	***
BV2	←	B	0.291	0.043	6.736	***
AV5	←	A	1			

*** Statistically significant at *p* < 0.001 level.

**Table 5 foods-13-00125-t005:** Path coefficients, S. E. values, C and Results of the SEM model.

Hypothesis				Estimate	S.E.	C.R.	P	Label
2	A	←	K	0.063	0.028	2.304	0.021	Accepted
3	SN	←	K	0.407	0.074	5.505	0.000	Accepted
4	PBC	←	K	0.044	0.047	0.938	0.348	Not accepted
5	I	←	K	0.406	0.063	6.461	0.000	Accepted
6	I	←	A	−0.018	0.102	−0.172	0.863	Not accepted
7	I	←	SN	0.344	0.038	9.057	0.000	Accepted
8	I	←	PBC	0.088	0.06	1.475	0.140	Not accepted
9	B	←	I	0.077	0.029	2.62	0.009	Accepted

## Data Availability

The data presented in this study are available on request from the corresponding author. The data are not publicly available due to due to privacy limitations concerning the use of personal information.

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
