# Peer review of "Fruit and Vegetable Consumption during the COVID-19 Lockdown in Serbia: An Online Survey"

_foods, 2023, doi:10.3390/foods13010125_

Round 1

Reviewer 1 Report

Comments and Suggestions for Authors

The manuscript addresses an interesting and valid research question. However, I have some suggestions to reshape the paper.

-          The abstract is too theoretical; instead of describing TPB, the results and the practical implications should be highlighted

-          in the introduction, I suggest following a different perspective: first describing the F&V consumption (in general and in Serbia) and then describing the effects of the pandemic (in food consumption and in F&V consumption, in general, and in Serbia)

-          In the literature review, I miss recent publications on European F&V consumption (e.g.: https://doi.org/10.1016/j.jafr.2023.100883 )

-          Regarding the sample, there is no validation of the data collection (particularly, why this convenient sampling method was followed). Was there any screening question?

-          In addition, no reference to whether ethical statements were available

-          For Table1 I suggest having the data for the total Serbian population, where it is applicable

-          Table2 might be moved to the Appendix

-          I miss a dedicated section for the practical implications in the Discussion/Conclusion

-          Also, there is no section for limitations, which is highly required, considering the sampling method

Some technical notes:

-          some references in the text are used not correctly (e.g., [18] )

-          some references in the Bibliography are incorrect (e.g., 2)

Author Response

The authors would like to thank the reviewers for their constructive suggestions that help us improve the manuscript. All changes made can be found in the manuscript via track changes option.

Reviewer 1

The manuscript addresses an interesting and valid research question. However, I have some suggestions to reshape the paper.

AUTHORS: Thank you very much for the possitive attitude toward our investigation.

-          The abstract is too theoretical; instead of describing TPB, the results and the practical implications should be highlighted

AUTHORS: Corrections were made according to the suggestions.

-          in the introduction, I suggest following a different perspective: first describing the F&V consumption (in general and in Serbia) and then describing the effects of the pandemic (in food consumption and in F&V consumption, in general, and in Serbia)

AUTHORS: Thank you very much for this suggestion, the text was rearranged according to the Reviewer’s opinion.

-          In the literature review, I miss recent publications on European F&V consumption (e.g.: https://doi.org/10.1016/j.jafr.2023.100883 )

AUTHORS: The reference was added.

-          Regarding the sample, there is no validation of the data collection (particularly, why this convenient sampling method was followed). Was there any screening question?

AUTHORS: As mentioned in the text, The sample consisted of 27.8% males and 72.2% females (Table 1). This disproportion can be explained by the fact that women are more engaged and influential on pur-chasing and creating family diet [21]. Also, the proportion of respondents aged between 25 and 45 years (63.0%), and of residents of big cities (59.7%) compared to those from villages or small towns around the country was dominant. Moreover, 69.7% of respondents had elementary or high school education, whereas only 30.3% had college degree. Also, 75.2% were employed while 76.8% had more than the average monthly income of approximately 500 €. Analysis of the family structure showed that 63.0% of respondents have 3 or more family members. It was an anonymous screening investigation designed to investigate fruit and vegetable consumption during the COVID-19 lock-down in Serbia.

-          In addition, no reference to whether ethical statements were available

AUTHORS: the following statement was added in the manuscript:

Informed Consent Statement: Informed consent was obtained from all subjects involved in the study. Written informed consent has been obtained from the patient(s) to publish this paper.

-          For Table1 I suggest having the data for the total Serbian population, where it is applicable

AUTHORS: Thank you for the suggestion. As you can see, this table refers to the socio-demographic characteristics of respondents in Serbia who participated in the research, and these data are important for understanding the obtained data. On the other hand, having data for the total population in Serbia is not fully applicable and was also outside the scope of this research, and therefore was not included in the study. Nevertheless, if the reviewer insists on having data for total Serbian population, we will add the required data where applicable.

-          Table2 might be moved to the Appendix

AUTHORS: Thank you for the suggestion. However, the authors strongly believe that moving Table 2 from the text to the appendix will disrupt the flow of the manuscript due to its content. This table presents an overview of the examined TPB constructs along with the responses and provides an overview which factors have an impact on consumers’ behavior. Nevertheless, if the reviewer insists on moving Table 2 in the Appendix, we will do as necessary.

-          I miss a dedicated section for the practical implications in the Discussion/Conclusion

AUTHORS:

The following lines were added to the text:

The practical use of this study was to examine the statistical viability of the hy-potheses stated in Table 5, to be used for similar purposes to predict and understand human behavior and habits in unexpected circumstances. However, it is important to note that the main limitation of the study may be the sample structure, since most of the participants were women. Therefore, in future studies, where possible and applicable, this factor should be more balanced.

Additionally, a significant contribution of this study lies in applying similar ana-lyzes in other countries of the region, especially those in the Mediterranean area, since consumers have similar attitudes and habits towards F&V consumption.

-          Also, there is no section for limitations, which is highly required, considering the sampling method

AUTHORS: As mentioned in the text, The sample consisted of 27.8% males and 72.2% females (Table 1). This disproportion can be explained by the fact that women are more engaged and influential on pur-chasing and creating family diet [21]. Also, the proportion of respondents aged between 25 and 45 years (63.0%), and of residents of big cities (59.7%) compared to those from villages or small towns around the country was dominant. Moreover, 69.7% of respondents had elementary or high school education, whereas only 30.3% had college degree. Also, 75.2% were employed while 76.8% had more than the average monthly income of approximately 500 €. Analysis of the family structure showed that 63.0% of respondents have 3 or more family members. It was an anonymous screening investigation designed to investigate fruit and vegetable consumption during the COVID-19 lock-down in Serbia.

Some technical notes:

-          some references in the text are used not correctly (e.g., [18] )

AUTHORS: corrections were made according to the reviewer’s suggestions

-          some references in the Bibliography are incorrect (e.g., 2)

AUTHORS: corrections were made according to the reviewer’s suggestions

The authors would like to thank the reviewers for their constructive suggestions that help us improve the manuscript. All changes made can be found in the manuscript via track changes option.

Reviewer 1

The manuscript addresses an interesting and valid research question. However, I have some suggestions to reshape the paper.

AUTHORS: Thank you very much for the possitive attitude toward our investigation.

-          The abstract is too theoretical; instead of describing TPB, the results and the practical implications should be highlighted

AUTHORS: Corrections were made according to the suggestions.

-          in the introduction, I suggest following a different perspective: first describing the F&V consumption (in general and in Serbia) and then describing the effects of the pandemic (in food consumption and in F&V consumption, in general, and in Serbia)

AUTHORS: Thank you very much for this suggestion, the text was rearranged according to the Reviewer’s opinion.

-          In the literature review, I miss recent publications on European F&V consumption (e.g.: https://doi.org/10.1016/j.jafr.2023.100883 )

AUTHORS: The reference was added.

-          Regarding the sample, there is no validation of the data collection (particularly, why this convenient sampling method was followed). Was there any screening question?

AUTHORS: As mentioned in the text, The sample consisted of 27.8% males and 72.2% females (Table 1). This disproportion can be explained by the fact that women are more engaged and influential on pur-chasing and creating family diet [21]. Also, the proportion of respondents aged between 25 and 45 years (63.0%), and of residents of big cities (59.7%) compared to those from villages or small towns around the country was dominant. Moreover, 69.7% of respondents had elementary or high school education, whereas only 30.3% had college degree. Also, 75.2% were employed while 76.8% had more than the average monthly income of approximately 500 €. Analysis of the family structure showed that 63.0% of respondents have 3 or more family members. It was an anonymous screening investigation designed to investigate fruit and vegetable consumption during the COVID-19 lock-down in Serbia.

-          In addition, no reference to whether ethical statements were available

AUTHORS: the following statement was added in the manuscript:

Informed Consent Statement: Informed consent was obtained from all subjects involved in the study. Written informed consent has been obtained from the patient(s) to publish this paper.

-          For Table1 I suggest having the data for the total Serbian population, where it is applicable

AUTHORS: Thank you for the suggestion. As you can see, this table refers to the socio-demographic characteristics of respondents in Serbia who participated in the research, and these data are important for understanding the obtained data. On the other hand, having data for the total population in Serbia is not fully applicable and was also outside the scope of this research, and therefore was not included in the study. Nevertheless, if the reviewer insists on having data for total Serbian population, we will add the required data where applicable.

-          Table2 might be moved to the Appendix

AUTHORS: Thank you for the suggestion. However, the authors strongly believe that moving Table 2 from the text to the appendix will disrupt the flow of the manuscript due to its content. This table presents an overview of the examined TPB constructs along with the responses and provides an overview which factors have an impact on consumers’ behavior. Nevertheless, if the reviewer insists on moving Table 2 in the Appendix, we will do as necessary.

-          I miss a dedicated section for the practical implications in the Discussion/Conclusion

AUTHORS:

The following lines were added to the text:

The practical use of this study was to examine the statistical viability of the hy-potheses stated in Table 5, to be used for similar purposes to predict and understand human behavior and habits in unexpected circumstances. However, it is important to note that the main limitation of the study may be the sample structure, since most of the participants were women. Therefore, in future studies, where possible and applicable, this factor should be more balanced.

Additionally, a significant contribution of this study lies in applying similar ana-lyzes in other countries of the region, especially those in the Mediterranean area, since consumers have similar attitudes and habits towards F&V consumption.

-          Also, there is no section for limitations, which is highly required, considering the sampling method

AUTHORS: As mentioned in the text, The sample consisted of 27.8% males and 72.2% females (Table 1). This disproportion can be explained by the fact that women are more engaged and influential on pur-chasing and creating family diet [21]. Also, the proportion of respondents aged between 25 and 45 years (63.0%), and of residents of big cities (59.7%) compared to those from villages or small towns around the country was dominant. Moreover, 69.7% of respondents had elementary or high school education, whereas only 30.3% had college degree. Also, 75.2% were employed while 76.8% had more than the average monthly income of approximately 500 €. Analysis of the family structure showed that 63.0% of respondents have 3 or more family members. It was an anonymous screening investigation designed to investigate fruit and vegetable consumption during the COVID-19 lock-down in Serbia.

Some technical notes:

-          some references in the text are used not correctly (e.g., [18] )

AUTHORS: corrections were made according to the reviewer’s suggestions

-          some references in the Bibliography are incorrect (e.g., 2)

AUTHORS: corrections were made according to the reviewer’s suggestions

Reviewer 2 Report

Comments and Suggestions for Authors

Although the topic is relevant and the approach is largely sound, there are some areas that need clarification, expansion, or re-examination.

1. The aim is missing from the summary.

2. Methodological issues:

The sampling method is missing from the description of the sample. How was the sample size determined? How does your sample match the general population in Serbia?

3. Discussion and conclusion:

Please improve the discussion on how the results can be used in other countries.

Provide implications for practise and research, and discuss the limitations of the study.

Author Response

The authors would like to thank the reviewers for their constructive suggestions that help us improve the manuscript. All changes made can be found in the manuscript via track changes option.

Reviewer 2

Although the topic is relevant and the approach is largely sound, there are some areas that need clarification, expansion, or re-examination.

AUTHORS: Thank you very much for the possitive attitude toward our investigation.

  1. The aim is missing from the summary.

AUTHORS: Thank you for this observation. A few lines of text was added in the Introduction section:

Having in mind contemporary studies regarding F&V consumption in Europe [17], the primary objective of the present research was to delve deeper into the intricacies of F&V consumption during COVID-19 and its correlation with the behavior of the participants in Serbia.

The initial hypothesis (H1) proposes that F&V consumption involves a complex inter-play of multiple factors. Subsequent hypotheses suggest that the knowledge (K) affects various elements, namely: (H2) overall attention toward F&V consumption (A); (H3) subjec-tive norm (SN); (H4), and personal behavior control (PBC). Furthermore, the intention of F&V consumption (I) is expected to be influenced by the previous knowledge (K), H5, attitude to consume F&V, H6, subjective norm H7 and personal behavior control, H8. The last hypothesis (H9) is regarding the influence of intention toward behavior of F&V consumption.

  1. Methodological issues:

The sampling method is missing from the description of the sample. How was the sample size determined? How does your sample match the general population in Serbia?

AUTHORS: According to text, the sample size was tested according to the calculation: http://danielsoper.com/statisticalc/calculator.aspx?id=89 and it was confirmed that the minimum sample should be 256 survey members. Additionally exploratory factor analy-sis (EFA) was used to test the adequacy of sampling through Kaiser-Mayer-Olkin measure of sampling (KMO≥0.5) [24, 25], while the strength of the relationship among variables was assessed through Bartlett’s test of sphericity where the value less than 0.05 indicates that these data should not be acceptable for further analysis [26].

  1. Discussion and conclusion:

Please improve the discussion on how the results can be used in other countries.

AUTHORS: The correction were made according to the reviewer’s suggestions and the following lines were added to the text:

Additionally, a significant contribution of this study lies in applying similar ana-lyzes in other countries of the region, especially those in the Mediterranean area, since consumers have similar attitudes and habits towards F&V consumption.

Provide implications for practise and research, and discuss the limitations of the study.

AUTHORS: The correction were made according to the reviewer’s suggestions and the following lines were added to the text:

The practical use of this study was to examine the statistical viability of the hy-potheses stated in Table 5, to be used for similar purposes to predict and understand human behavior and habits in unexpected circumstances. However, it is important to note that the main limitation of the study may be the sample structure, since most of the participants were women. Therefore, in future studies, where possible and applicable, this factor should be more balanced.

Additionally, a significant contribution of this study lies in applying similar ana-lyzes in other countries of the region, especially those in the Mediterranean area, since consumers have similar attitudes and habits towards F&V consumption.

Round 2

Reviewer 1 Report

Comments and Suggestions for Authors

Thank you, the authors have addressed most of my concerns.

Regarding a section for limitations I meant a textual explanation due to the data collection (e.g., convenient sampling) and the methodology applied. These should be clearly highlighted.

Author Response

Reviewer 1

Thank you, the authors have addressed most of my concerns.

ANSWER: The Authors would like to thank the Reviewer for a quick and professional review. The Authors believe that the actual paper would satisfy the scientific community and that it is going to be interesting enough for publishing in the Journal.

Regarding a section for limitations I meant a textual explanation due to the data collection (e.g., convenient sampling) and the methodology applied. These should be clearly highlighted.

ANSWER: Thank you very much for this question. Generally, sampling method in survey analysis could implement different limitations that might influence the accuracy and representativeness of the collected data. Several common limitations may include:

  1. Sampling Bias: The chosen sample might not be representative of the entire population due to biases in selection method. For instance, if a survey targets only a specific age group or geographic region, it may not accurately reflect the opinions of the entire population.
  2. Non-response Bias: Not everyone contacted for a survey participates. If certain groups are more likely to decline or not respond, this can skew the results and make them less representative.
  3. Small Sample Size: A small sample size might not capture the diversity of opinions within the population, leading to less reliable results and wider margins of error.
  4. Sampling Frame Issues: Inaccurate or incomplete sampling frames, which are lists used to identify the population to be sampled, can lead to exclusion or over-representation of certain groups.
  5. Sampling Method Errors: Errors in the method of sampling, such as using convenience sampling instead of random sampling, can introduce biases and affect the validity of the survey results.
  6. Inability to Generalize Findings: Sometimes, due to limitations in the sampling method, the findings of a survey cannot be generalized to the entire population, especially if certain groups are underrepresented or excluded.

Addressing these limitations often involves employing suitable sampling techniques, ensuring the representation of a diverse sample, minimizing non-response, and maintaining transparency regarding the data collection methods used.

According to this, the authors added fallowing paragraph into the manuscript:

However, it is important to note that the main limitation is related to sampling and sample structure, since the majority of the participants were women. Therefore, in future studies, where possible and applicable, there should be a more balanced gender representation. Generally, sampling limitations in survey analysis include biases (sampling, non-response), small sample sizes, inaccuracies in sampling frames/methods, and challenges in generalizing findings. Addressing these concerns involves using appropriate techniques, ensuring diversity, minimizing non-response, and maintaining transparency in data collection.
